# Long-Term Fairness Without Utility Deterioration

## Abstract

In fair machine learning, the trade-off between fairness and utility has been predominantly studied in static classification settings, neglecting concerns for long-term learning environments where the population distribution may vary due to the deployment of model policies. This work investigates whether zero utility deterioration can be achieved in the long run. We introduce a Markov decision process (MDP) to formulate the interplay between model decisions and population distribution shifts. A key technical contribution is identifying a sufficient and necessary condition under which a model policy achieving long-term fairness does not compromise utility. Inspired by this condition, we propose effective reward functions that can be combined with online reinforcement learning algorithms, allowing the classifier to accommodate dynamic control objectives such as inducing population adaptations to maximize fairness without sacrificing model performance. Experiments on both synthetic and real-world datasets suggest the effectiveness of the proposed reinforcement learning framework in the long run and drive a classifier-population system toward a desirable equilibrium where the identified condition is met.

## 1 Introduction

The deployment of machine learning models carries a critical need to eliminate algorithmic discrimination in numerous high-stakes real-world applications, including recommender systems (Li et al., 2023), hiring processes (Makhlouf et al., 2020), and targeted advertising (Papakyriakopoulos et al., 2022). For example, facial recognition models have been reported to show deficiencies in recognizing individuals with darker skin tone (Buolamwini & Gebru, 2018), while recruitment models display bias favoring male candidates over equally qualified female applicants (Kiritchenko & Mohammad, 2018). In response, researchers have introduced a number of fairness notions, such as Demographic Parity (`DP`) (Dwork et al., 2012), Equality of Opportunity (`EqOpt`) (Hardt et al., 2016), among others (Dwork et al., 2012; Kusner et al., 2017). However, enforcing the fairness constraints will inevitably worsen the prediction performance of the decision models, resulting in a trade-off phenomenon that has been both theoretically characterized and empirically observed in the literature (Menon & Williamson, 2018; Zhao & Gordon, 2022; Chen et al., 2018; Zhang et al., 2020). The decline in utility that occurs when fairness constraints are intervened, compared to an unconstrained model, is referred to as *deterioration*.

Previous works have adequately studied the utility deterioration in static learning environments. For example, Dutta et al. (2020) demonstrated that it is impossible to achieve fairness without sacrificing the accuracy unless the degree of "separability" within the class conditional distribution among two groups is equal. Rodolfa et al. (2021) proposed that post-processing with a group-specific threshold score can improve fairness with modest deterioration. Wick et al. (2019) characterized the conditions under which fairness and utility mutually benefit each other. Nevertheless, it is crucial to recognize that machine learning policies and populations can mutually adapt to each other, leading to a shift in the underlying data distribution and a changing environment for the policymaker. This dynamic interplay poses a fundamental research question in the pursuit of long-term fairness: *can we achieve fairness without compromising utility deterioration in a dynamic environment?*

Our work attempts to address the utility deterioration in the long run. When considering the long-term impact of algorithmic decisions, the underlying population dynamically interacts with the decisions made

by the decision policies. On one hand, as the population distribution is dynamic and responsive to these decisions, there is a plausible prospect that employing an apt policy could gradually steer the population distribution toward a desirable trajectory (e.g., balanced data distributions across groups) that will ultimately eliminate the fairness-utility trade-off. On the other hand, a core practical difficulty posed in the proposed problem is the potentially unknown dynamics of the system under control. Even with known dynamics, identifying such a desired policy remains a formidable challenge. For example, Zhang et al. (2020) examined the long-term impact of a myopic policy—one that optimizes utility in the short term while adhering to certain fairness notions —and found that it failed to achieve desirable socio-economic outcomes e.g., high qualification rates for both groups). Similarly, Tang et al. (2022) demonstrated that even a perfect predictor, achieving zero error rate at each time step, falls short of achieving long-term fairness goals. We summarize our contributions as follows:

*The concept of utility deterioration.* For the observed data, we define the deterioration as the difference between the optimal utility value of a constrained optimization problem (i.e., maximizing utility subject to a fairness constraint) and that of an unconstrained optimization problem (i.e., maximizing utility only). We consider a general notion of utility, which can model the objective quantity considered in some previous works as special cases (e.g. the *reward* (Zhang et al., 2020) and *accuracy* (Dutta et al., 2020)).

*A necessary and sufficient condition of zero deterioration.* We show that there is no utility deterioration if and only if qualification rates of different demographic groups are equal, under two realistic assumptions. We provide the full proof in the supplementary material.

*An intervention to prevent the deterioration in the long run.* We use Markov Decision Process (MDP) (Puterman, 1994) to model the interplay between the algorithmic decisions and the underlying population in a sequential decision-making setting. Guided by the identified condition, we propose effective reward functions for online reinforcement learning (RL) algorithms. We show that our RL formulation of long-term fairness-utility trade-off allows an agent to learn to steer the system towards a desirable equilibrium without utility deterioration.

*Simulations on synthetic and real-world examples.* We conduct extensive empirical evaluations on both synthetic and real-world datasets. Experiments show that the proposed method is effective at achieving zero deterioration in the long run, and flexible to incorporate other long-term goals. We also identify and discuss the failure cases of our method.

## 2 Related Work

### 2.1 Fairness-utility trade-off

The fundamental question of fairness-utility trade-off has been studied predominantly in static settings and with a narrow utility notion of *accuracy*. Utility is inherently a broader concept than accuracy, as it encompasses various factors. The existence of fairness-accuracy trade-off has been observed empirically supported by prior studies (Žliobaitė, 2015; Zhao et al., 2019; Corbett-Davies et al., 2017), which noted that the pursuit of fairness often comes at the expense of accuracy. Building on this motivation, (Dutta et al., 2020; Zhao & Gordon, 2022; Xian et al., 2023) have delved into precisely characterizing the relationship between fairness and accuracy. For example, Zhao & Gordon (2022) specifically investigated the trade-off between demographic parity (DP) and classification accuracy. Under the assumption of a noiseless Bayes classifier and binary classes, they concluded that the inherent trade-off does not exist if and only if the base rates of two demographic groups are equal. In cases where this equality does not hold, any fair classifier must inevitably contend with a lower bound on classification error, proportionally linked to the disparity in base rates between the two demographic groups. Xian et al. (2023) further expanded on this by showing that the classification error rate for any DP fair classifier can be equated to the solution of a relaxed Wasserstein-barycenter problem. Importantly, this result applies without relying on the assumptions of a noiseless classifier and binary classes made in the earlier work by Zhao & Gordon (2022). Dutta et al. (2020) took a slightly different approach by examining an approximate measure of the trade-off between classification accuracy and equalized opportunity (EqOpt) fairness. They established that the trade-off is inevitable unless the degree of *separability* within the class-conditional distribution among two groups is equivalent.

## 2.2 Long-term fairness via reinforcement learning

While the application of fairness constraints is a common strategy to mitigate discrimination and biases in static scenarios, it can lead to adverse consequences for the population's well-being in the long term, as highlighted by the literature (Liu et al., 2018; Zhang et al., 2020; Tang et al., 2022; D'Amour et al., 2020; Guldogan et al., 2023; Perdomo et al., 2020). The fundamental reason behind this failure is that static fair classifiers lack an understanding of the temporal evolution of the data distribution, which can result in unpredictable outcomes for the population's well-being over an extended duration. Given that long-term fairness issues entail sequential decision-making, they naturally fit within the framework of RL. Wen et al. (2021); Yu et al. (2022); Xu et al. (2024) pioneered in proposing the use of RL to address long-term fairness concerns. Subsequently, Yin et al. (2023) and Deng et al. (2023) independently devised RL algorithms with step-wise fairness constraints, demonstrating that RL algorithms can discover policies capable of guiding a responsive population towards states characterized by both higher accuracy and fairness. Yin et al. (2023) considered a replicator dynamic (Raab & Liu, 2021), while Deng et al. (2023) employed a partially observed MDP to define the data dynamics, akin to the approach in (Zhang et al., 2020).

Notably, our work differs from prior research in its overarching objective: we aim to understand the fairness-utility trade-off in the dynamic setting where the algorithmic decisions and the underlying population mutually adapt to one another. More importantly, we seek to identify the sufficient and necessary conditions and effective interventions to help eliminate the utility-fairness trade-off in the long run. Consequently, we depart from the conventional approach of utilizing immediate accuracy and fairness as rewards. Instead, we derive our reward signal from the identified conditions for achieving a zero trade-off. The outcome is an online RL framework that considers potential unknown dynamics and steers the population to a desirable socioeconomic status.

## 3 Problem Formulation

We consider a standard classification task with binary sensitive attributes, denoted by $G \in \{a, b\}$. The fraction of the population in protected group $G = g$ is denoted as $p_g$, i.e., $p_a = \mathbb{P}(G = a)$ and $p_b = \mathbb{P}(G = b) = 1 - p_a$. Each individual in these groups possesses an observed feature $X$ and binary label $Y \in \{0, 1\}$. We denote the *qualification rate* of a group $s$ as $\alpha_s$, which is defined as the probability of an individual in that group having labels $Y = 1$, i.e.

$$\alpha_g := \mathbb{P}(Y = 1 \mid G = g). \tag{1}$$

To shorten the notations, we also represent the group distribution as

$$\rho_g(x) = \mathbb{P}(X = x \mid G = g). \tag{2}$$

We assume that the model has access to the sensitive attributes when making predictions. Let $\hat{Y}$ denote the prediction of the classifier, we denote the groupwise *policy* of the decision model as $\pi_g(x) := \mathbb{P}(\hat{Y} = 1 \mid X = x, G = g)$. For ease of notation, we use $\pi = (\pi_a, \pi_b)$ to denote the aggregated model policy. Under the long-term setting, each random variable defined above will (by default) change over time following certain dynamics, we omit the time subscript $t$ here and defer the formal definition of the dynamic for the sake of notation simplicity.

Throughout this work, we focus on two standard group fairness metrics:

**Definition 1** (Demographic parity (Dwork et al., 2012)). A model policy satisfies *demographic parity* if the selection rates are equal across groups, *i.e.*,

$$\mathbb{P}(\hat{Y} = 1 \mid G = a) = \mathbb{P}(\hat{Y} = 1 \mid G = b). \tag{DP}$$

**Definition 2** (Equality of opportunity (Hardt et al., 2016)). A model policy satisfies *equality of opportunity* if the true positive rates are equal across groups, *i.e.*,

$$\mathbb{P}(\hat{Y} = 1 \mid Y = 1, G = a) = \mathbb{P}(\hat{Y} = 1 \mid Y = 1, G = b). \tag{EqOpt}$$

We then define a $2 \times 2$ cost matrix $C$, where each entry $c_{ij}$ captures the loss or gain that the decision-maker can experience for various combinations of predictions $\hat{Y} = i$ and ground truth labels $Y = j$. The utility of a model policy $\pi$ is defined as the expectation of cost:

$$\text{Util}(\pi) := \sum_{i,j} c_{ij} \mathbb{P}(\hat{Y} = i, Y = j). \tag{3}$$

We remark that the utility defined in Eqn (3) is also known as Bayes risk. *Accuracy* can be instantiated by assigning $c_{00} = c_{11} = 1, c_{01} = c_{10} = 0$. Given its flexibility, utility is more desirable than the *accuracy* in many decision-making scenarios.

## 4 When Is There No Utility Deterioration?

Our central pursuit is to investigate whether there exists a trade-off between fairness and utility. In particular, we first formulate the definition of utility deterioration, which quantifies the sacrifice in utility that must be incurred to attain fairness. Our main result (Theorem 3) shows that utility deterioration is completely avoided (i.e., there is no trade-off between fairness and utility) *if and only if* the qualification rates across different demographic groups are identical.

Previous works (Zhao & Gordon, 2022; Dutta et al., 2020) measured the trade-off between fairness and accuracy using the *accuracy* of the optimal fair classifier, assuming the optimal unconstrained classifier (Bayes Classifier) makes the perfect prediction. Likewise, we define utility deterioration as the (absolute) difference between the optimal utility value of a fairness-constrained optimization problem (*i.e.*, maximizing utility subject to a fairness constraint) and that of an unconstrained optimization problem (*i.e.*, maximizing utility only).

**Definition 3** (Utility deterioration). Let $\pi^*$ denote the optimal solution to the unconstrained optimization problem

$$\pi^* = \max_\pi \text{Util}(\pi), \tag{4}$$

and $\pi^\diamond$ denote the optimal solution to the fairness-constrained optimization problem

$$\pi^\diamond = \max_\pi \text{Util}(\pi) \qquad \text{subject to} \quad \text{DP or EqOpt holds.} \tag{5}$$

The utility deterioration is defined as

$$\Delta := \text{Util}(\pi^*) - \text{Util}(\pi^\diamond). \tag{6}$$

$\Delta$ is non-negative due to the optimality of $\pi^*$, and there will be no deterioration if and only if $\text{Util}(\pi^*) = \text{Util}(\pi^\diamond)$. However, solving the optimization problems defined in Eqn (5) is challenging in practice due to the enforcement of nonconvex and nondifferentiable parity constraints. A series of prior works (Dwork et al., 2012; Hardt et al., 2016; Kusner et al., 2017; Dutta et al., 2020) suggested replacing them with their empirical counterpart estimated from data and proposed different ways to train a classifier that satisfies group fairness. Although effective, these methods can only at best give an empirical estimation. An exact solution is still required to identify the condition of $\Delta = 0$.

It is worth noting that it always suffices to develop a model that takes multivariate features as input and maps them to a scalar likelihood score. Taking the loan application example, credit bureaus assess an individual's credit score based on their profiles and credit reports. Applicants with lower credit scores typically have a lower probability of default of payments ($Y = 0$) compared to applicants with higher credit scores. For the sake of simplicity, we assume the scalar feature $X \in R$ and a mild monotonicity condition, allowing the decision-maker to set a threshold value for making a decision based on that score.

**Definition 4** (Monotone likelihood ratio). We say that the feature $X$ is *well-behaved* if for any group $g$, the distributions $\mathbb{P}(X = x \mid Y = 1, G = g)$ and $\mathbb{P}(X = x \mid Y = 0, G = g)$ have the monotone likelihood ratio property. That is,

$$\forall g \in \{a, b\}, \qquad \frac{\partial}{\partial x} \frac{\mathbb{P}(X = x \mid Y = 1, G = g)}{\mathbb{P}(X = x \mid Y = 0, G = g)} \geq 0. \tag{7}$$

Note that the above condition is relatively mild, as the likelihood ratio is always monotonic for a Bayes-optimal classifier. Then we are ready to show that the threshold classifiers are optimal solutions to the above optimization problems in Eqn (4) and Eqn (5).

**Lemma 1.** *Assuming that the feature is well-behaved, there exist two threshold pairs $(\mu_a, \mu_b)$, $(\nu_a, \nu_b)$, such that*

$$\pi_a^*(x) := \mathbb{1}(x \geq \mu_a), \quad \pi_b^*(x) := \mathbb{1}(x \geq \mu_b) \tag{8}$$

*is the optimal policy pair for the unconstrained optimization problem Eqn (4); and*

$$\pi_a^\diamond(x) := \mathbb{1}(x \geq \nu_a), \quad \pi_a^\diamond(x) := \mathbb{1}(x \geq \nu_b) \tag{9}$$

*is the optimal policy pair for the fairness-constrained optimization problem defined in Eqn (5).*

Let $\gamma_g(x) = \mathbb{P}(Y = 1 | X = x, G = g)$ denote the real qualification rate. The utility for a threshold classifier $\pi(x) = \mathbb{1}(x \geq \mu)$ can be explicitly decomposed as the sum of incurred cost for predicted negative examples ($\hat{Y} = 0$) and that for predicted positive examples ($\hat{Y} = 1$). We define the incurred cost $U_g^+$ and $U_g^-$ as

$$U_g^+ = \int_{\mu_g}^{+\infty} \left( c_{10}(1 - \gamma_g(x)) + c_{11}\gamma_g(x) \right) \rho_g(x) dx \tag{10}$$

$$U_g^- = \int_{-\infty}^{\mu_g} \left( c_{00}(1 - \gamma_g(x)) + c_{01}\gamma_g(x) \right) \rho_g(x) dx \tag{11}$$

By the law of total probability, we have

$$\text{Util}(\pi) = \sum_g p_g (U_g^- + U_g^+). \tag{12}$$

The above expression enables us to derive the optimal threshold policies.

**Lemma 2** (Optimal policy). *For any group $g$, the optimal threshold $\mu_g$ for unconstrained optimization problem satisfies $\gamma_g(\mu_g) = \gamma^*$, where*

$$\gamma^* = \frac{c_{00} - c_{10}}{c_{00} + c_{11} - c_{10} - c_{01}}. \tag{13}$$

*The optimal threshold pair $(\nu_a, \nu_b)$ for the constrained optimization problem satisfies*

$$\sum_g p_g \left( \gamma_g(\nu_g) - \gamma^* \right) = 0, \quad \text{or} \tag{DP constrained}$$

$$\sum_g p_g \left( 1 - \frac{\gamma^*}{\gamma_g(\nu_g)} \right) \mathbb{P}(Y = 1 \mid G = g) = 0 \tag{EqOpt constrained}$$

*where $p_g = \mathbb{P}(G = g)$ represents the population distribution.*

Lemma 2 implies that the qualification rate at the unconstrained threshold, $\gamma^*$, is a linear combination of the qualification rates at the corresponding fair thresholds. Given the optimal unconstrained threshold $\mu_g$ and the fair threshold $\nu_g$ for each group $g$, the incurred utility deterioration is

$$\Delta = \sum_g p_g \Delta_g, \tag{14}$$

where

$$\Delta_g = (c_{00} + c_{11} - c_{01} - c_{10}) \int_{\mu_g}^{\nu_g} (\gamma_g(x) - \gamma^*) \cdot \rho_g(x) dx.$$

Note that the above analysis still holds for multiple sensitive attributes. Next, we describe the separation assumption which is also used in (Raab & Liu, 2021).

**Definition 5** (Separation). We say that the feature $X$ is *well-separated* if it is conditional independent of the sensitive attribute $G$ given the true qualification $Y$, i.e. $X \perp\!\!\!\perp G \mid Y$. Specifically, for all values of $x \in R$ and $y \in \{0, 1\}$, $\mathbb{P}(X = x \mid Y = y, G = g)$ remains the same across different groups $g$.

The separation condition states that an individual's feature (e.g. credit score) does not encode any sensitive attribute (e.g. race) given their qualification status. This assumption is realistic in the sense that all of the non-sensitive information of an individual fully corresponds to their true qualification. Under such assumption, we can derive a necessary and sufficient condition for zero deterioration.

**Theorem 3.** *Suppose that the feature $X$ is both well-behaved and well-separated.*

1. *For `DP`, there is no utility deterioration if and only if the qualification rates of the two groups are equal, i.e. $\alpha_a = \alpha_b$, if the following equation does not admit other solutions for the specific choices of population distributions of $\mathbb{P}(X = x \mid Y = 1, G = g)$ and cost matrix $c_{ij}$:*

$$\int_{\gamma_a^{-1}(\gamma^*)}^{\infty} \rho_a(x)dx = \int_{\gamma_b^{-1}(\gamma^*)}^{\infty} \rho_b(x)dx. \tag{15}$$

2. *For `EqOpt`, there is no utility deterioration if and only if the qualification rates of the two groups are equal, i.e. $\alpha_a = \alpha_b$.*

## 5 How to Eliminate Utility Deterioration in the Long Term?

Theorem 3 indicates that the identical qualification rates across different groups sufficiently imply no utility deterioration. In a dynamic learning environment, the deployment of model policy can change the underlying population distribution over time in turn. In this section, we explore whether there exist specific interventions (which are instantiated by decisions) that can lead the population distribution to respond in a manner that progressively brings it closer to a regime where the qualification rates across groups become equal. With repeated interventions over time, the population distribution could ultimately reach an ideal regime with zero deterioration. To answer this question, we model this problem as a *Markov decision process* (MDP) Puterman (1994), and then naturally we can use an online RL algorithm to tackle the problem.

**MDP setup.** We consider a MDP environment described by a tuple $\langle \mathcal{S}, \mathcal{A}, \mathbb{P}_T, R \rangle$, where $\mathcal{S}$ is the set of environment states, $\mathcal{A}$ is the set of actions the agent can take, $\mathbb{P}_T$ is the state transition probability, $R$ is the reward function. Following Yin et al. (2023), we assume that $\mathbb{P}_T, R$ do not depend on time. The agent interacts with the environment as follows. The initial state is given by $s_0$. At each time step $t$, the agent observes a state $s_t$, and chooses an action specified by the policy $\pi_t : \mathcal{S} \to \mathcal{A}$. Then the agent receives a reward $r_t \sim R(s_t, a_t)$, finally, the environment evolves into a new state $s_{t+1} \sim \mathbb{P}_T(\cdot \mid s_t, a_t)$. In the long-term setting, a classification model makes predictions at each time step, and then the population distribution shifts in response to the prediction. We identify the population distribution, which is the joint probability distribution $\mathbb{P}_t(X, Y, G)$, as the state at time $t$. Throughout this section, we will use subscript $t$ to represent the feature $X_t$ and true qualification $Y_t$ of individuals at time step $t$.

We assume that the group distribution $\mathbb{P}(G = g)$ will not change in the transition process. We assume that the population dynamics is subject to a label shift, which means the data generation distribution $\mathbb{P}(X = x \mid Y = y, G = g)$ is independent of time $t$. Both assumptions are realistic in the real world decision-making process. Notice that the joint distribution can be decomposed as

$$\mathbb{P}(X = x, Y = y, G = g) = \mathbb{P}(Y = y \mid G = g) \cdot \mathbb{P}(X = x \mid Y = y, G = g)\mathbb{P}(G = g) \tag{16}$$

Denote the temporal qualification rate by

$$\boldsymbol{\alpha}_t^g := \begin{bmatrix} \mathbb{P}(Y = 0 \mid G = g, T = t) \\ \mathbb{P}(Y = 1 \mid G = g, T = t) \end{bmatrix} \tag{17}$$

Then the vector $\boldsymbol{\alpha}_t^g$ captures all the changes in the state and suffices to represent the state $s_t$ given the priors of $\mathbb{P}(X = x \mid Y = y, G = g)$ and $\mathbb{P}(G = g)$. We remark that while the qualification rates $\boldsymbol{\alpha}$ are not always

accessible to the policy maker in real-world problems, one can estimate the empirical distribution $\mathbb{P}(Y)$ from historical observations.

The action $\mathcal{A}$ is the set of classifiers $\mathbb{P}(\hat{Y} \mid X = x, G = g)$. For empirical consideration, we further constrain the action space to be the set of threshold policies: $\pi_g = \mathbb{1}(x \geq \theta_g)$, which essentially shrinks the search space from whole function space to a two-dimensional real space: $(\theta_a, \theta_b) \in \mathbb{R}^2$.[1]

**Modeling the dynamics.**   To model the evolution of the population distribution (equivalently, qualification rates), we follow Zhang et al. (2020) to specify the probability transition matrix $\boldsymbol{T}^g$ of an individual from group $G = g$ becomes qualified (i.e. $Y_{t+1} = 1$) after receiving decision/prediction $\hat{Y}_t$. That is, for all $y, \hat{y} \in \{0, 1\}$, we have

$$T_{y,\hat{y}}^g := \mathbb{P}(Y_{t+1} = 1 \mid Y_t = y, \hat{Y}_t = \hat{y}, G = g). \tag{18}$$

Then the transition of the state can be instantiated by the following dynamic for each group $g$:

$$\boldsymbol{\alpha}_{t+1}^g = \mathbb{E}_{X_t, Y_t, G=g} \left[ \boldsymbol{\alpha}_t^g \boldsymbol{T}^g \hat{\boldsymbol{Y}}_t^g \right], \tag{19}$$

where $\hat{\boldsymbol{Y}}_t^g = [1 - \pi_g(X_t), \pi_g(X_t)]^\intercal$ denotes the model prediction. The above evolution model is more general than using a specific form of dynamic, *e.g.*, additive dynamic (Liu et al., 2018) and multiplicative dynamic (Tang et al., 2022).

**Corollary 4.** *Under the above MDP setting, equalizing the qualification rates $\alpha$ at each time step is sufficient to prevent utility deterioration in the long run.*

When a demographic parity constraint is imposed on the model prediction, *i.e.* $\hat{\boldsymbol{Y}}_t^a = \hat{\boldsymbol{Y}}_t^b$, for all groups, the update dynamics become identical across groups. As a result, the group-wise qualification rates $\boldsymbol{\alpha}^g$ evolve under a common transition operator and will converge to the same stationary distribution at the limit. Therefore, all groups will eventually share the same qualification rate, and any disparity or deterioration present initially will asymptotically vanish.

**Reward function.**   We craft the intermediate reward at time $t$ using the absolute difference of qualification rate

$$R_{\texttt{base}}(s_t, a_t) := 1 - \frac{1}{\sqrt{2}} \|\boldsymbol{\alpha}_t^a - \boldsymbol{\alpha}_t^b\|. \tag{20}$$

The scale factor $\frac{1}{\sqrt{2}}$ is used for normalization to ensure that the range of the reward is between 0 and 1. Intuitively, using Eqn (20) as a reward function can encourage equal qualification rates between groups, thus achieving zero utility deterioration. However, we also need to consider the decision made at each time step to be (1) accurate with the 0-1 loss

$$\ell_{\texttt{acc}}(s_t, a_t) = \left| \mathbb{P}_{\substack{(X,Y,G)\sim s_t \\ \hat{Y} \sim a_t(X,G)}}(\hat{Y} \neq Y) \right|, \tag{21}$$

and (2) fair by penalizing the following fairness violations:

$$\ell_{\texttt{DP}}(s_t, a_t) = |\mathbb{P}(\hat{Y} = 1 \mid G = a) - \mathbb{P}(\hat{Y} = 1 \mid G = b)| \tag{22}$$

$$\ell_{\texttt{EqOpt}}(s_t, a_t) = |\mathbb{P}(\hat{Y} = 1 \mid Y = 1, G = a) - \mathbb{P}(\hat{Y} = 1 \mid Y = 1, G = b)| \tag{23}$$

The above loss function can be easily incorporated into the reward by inserting a regularization term to Eqn (20) as follows:

$$R(s_t, a_t) = R_{\texttt{base}}(s_t, a_t) - \lambda_1 \ell_{\texttt{acc}}(s_t, a_t) - \lambda_2 \ell_{\texttt{fair}}(s_t, a_t)) \tag{24}$$

where $\lambda_1 > 0$ and $\lambda_2 > 0$ are hyperparameters that can control the strength of accuracy and fairness regularizer, and $\ell_{\texttt{fair}}$ is one of the DP or EqOpt constraint as shown in Eqn (22) and Eqn (23). We note any RL algorithms can be applied to find the optimal policy so that the expected cumulative reward $\mathbb{E}(\sum_t r_t)$ is maximized, where the expectation is taken with respect to the historical observations.

---

[1]By Theorem 3, there exists a threshold policy that achieves maximum utility with or without fairness constraint, thus constraining action space to thresholds will not affect the best achievable instantaneous reward.

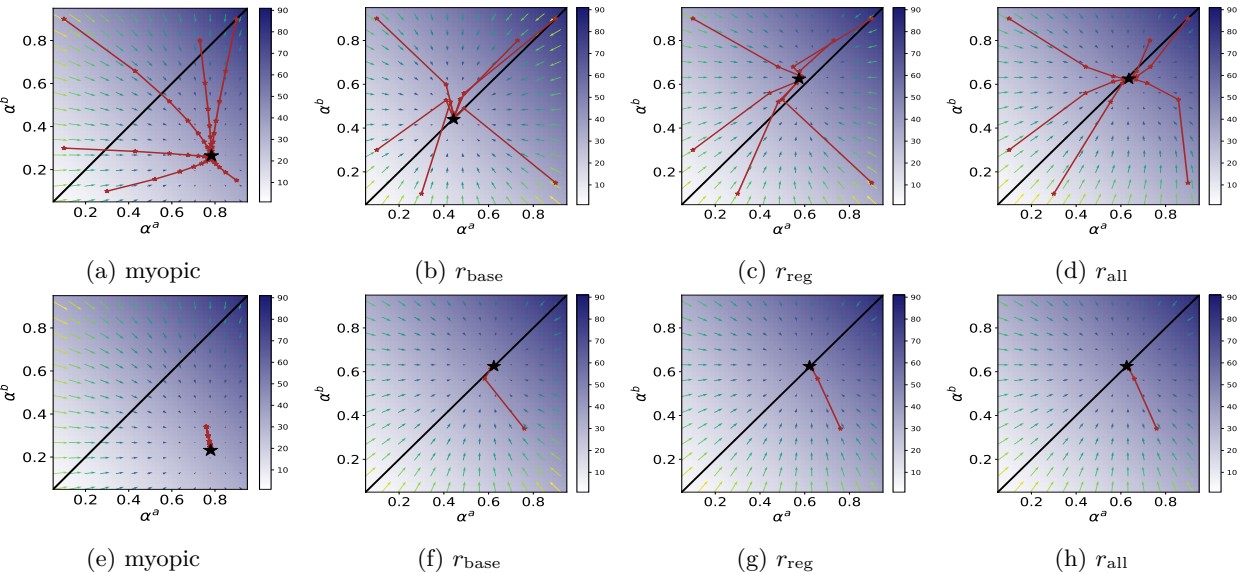

Figure 1: Trajectories for qualification rates with different initialization for synthetic Gaussian data (top) and the FICO dataset (bottom). In each subfigure for the synthetic data, six trajectories are plotted for 20 time steps with different initial qualification rates. The initial base rates for the FICO Score are estimated from the original data. The maximized utilities subject to fairness constraint `EqOpt` at every configuration of $(\alpha_a, \alpha_b)$ are computed and depicted using a heatmap. Qualification rates near the top-right corner achieve larger utility subject to `EqOpt`.

## 6 Experiment

### 6.1 Experimental Setup

**Datasets** We consider one synthetic dataset and two real-world datasets that are commonly used in the literature on long-term fairness (e.g. (Zhang et al., 2020; Tang et al., 2022; Liu et al., 2018)) in our study: the FICO dataset (Reserve, 2007) and the COMPAS (Angwin et al., 2016) dataset. We generate the synthetic data as follows: We set $X_t \mid Y_t = y, G = g \sim \mathcal{N}(\mu_y^s, (\sigma_y^s)^2)$. Specifically, to implement the separation (Definition 5), we set $\mu_1^a = \mu_1^b = 5, \mu_0^a = \mu_0^b = -5$. $\sigma_y^s = 5, \forall y, s$, it can be also verified that the feature is well-behaved (Definition 4). The transition probabilities are set to the same as Zhang et al. (2020): $T_{00}^a = 0.1, T_{01}^a = 0.5, T_{10}^a = 0.5, T_{11}^a = 0.7, T_{00}^b = 0.4, T_{01}^b = 0.5, T_{10}^b = 0.5, T_{11}^b = 0.9$.

The FICO score dataset (Reserve, 2007) contains credit score data from non-Hispanic White and Black cohorts. We use the pre-processed data by Hardt et al. (2016) and simulate the data-generating process according to the empirical distributions. Specifically, we consider Caucasian group ($G = a$) and African American group ($G = b$), with $\mathbb{P}(G = a) = 88\%, \mathbb{P}(G = b) = 12\%$. We compute the initial qualification rates from the original data and fit the feature distribution with the beta distribution. The transition probabilities are set to the same as the synthetic Gaussian dataset.

The COMPAS dataset (Angwin et al., 2016) is a high-dimensional recidivism prediction dataset. It has 10-dimensional feature $X$ and two demographic groups: Caucasian group ($G = a$) and African American group ($G = b$), with $\mathbb{P}(G = a) = 60\%, \mathbb{P}(G = b) = 40\%$. The initial qualification rates (recidivism rate) are calculated from raw data, which is $\alpha_a = 0.523, \alpha_b = 0.391$. To handle the high-dimensional feature, we first train an optimal classifier using a logistic regression model $\mathbb{P}(Y = 1 \mid X = x, G = g)$, which maps the high-dimensional feature to one-dimensional. Given a set of transition probabilities, the qualification rates change at each time step according to the dynamic defined in Eqn (19). Then we resample the dataset based on the updated qualification rate.

We will use the synthetic data and FICO data to present the primary results. Due to the high complexity of feature distribution, we use the high-dimensional COMPAS data for the discussion of the effect of altering transition probabilities.

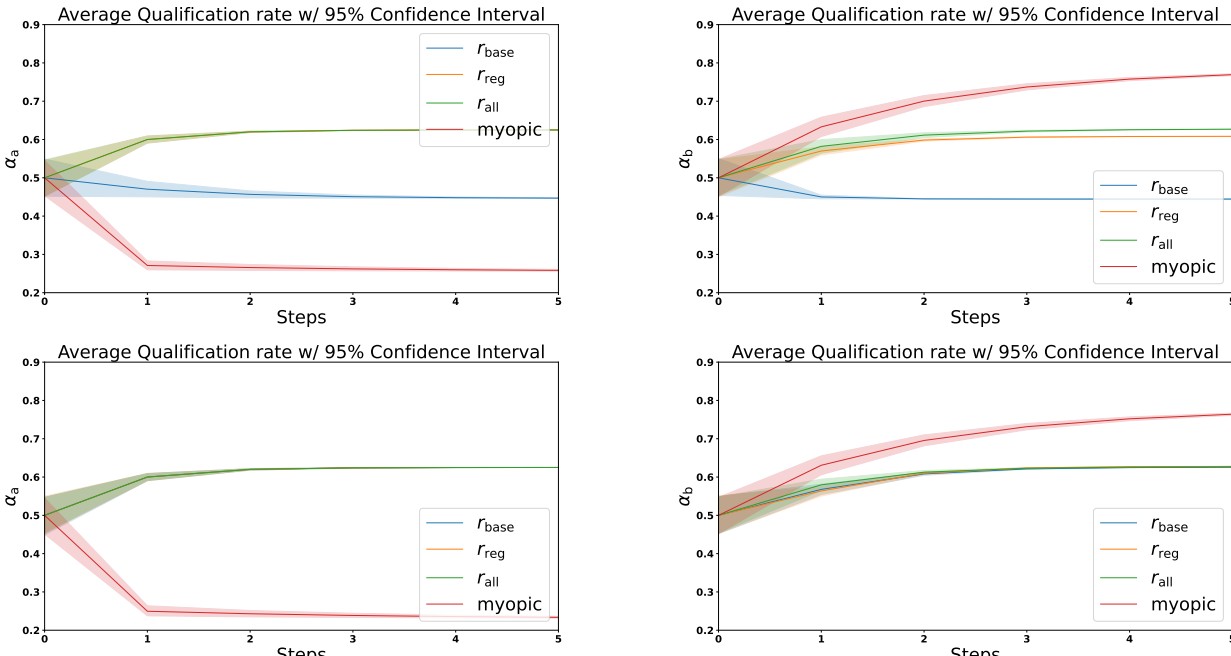

Figure 2: The qualification rate dynamics for group $a$ and group $b$ are on synthetic Gaussian data and the FICO dataset. From left to right: the first plot shows the qualification rate of group $a$ on the synthetic data, the second plot shows the qualification rate of group $b$ on the synthetic data, the third plot shows the qualification rate of group $a$ on the FICO dataset, and the rightmost plot shows the qualification rate of group $b$ on the FICO dataset.

**Models** We employ the "Stable-baselines 3" Raffin et al. (2021) implementation of Proximal Policy Optimization (PPO) Schulman et al. (2017) (can be replaced with any RL framework) for optimization. We compare the three different rewards:

- $r_{\texttt{base}}$ is the intermediate reward of $R_{\texttt{base}}(s_t, a_t)$, which only equalizes the qualification rate by setting $\lambda_1 = \lambda_2 = 0$.

- $r_{\texttt{reg}} = R_{\texttt{base}}(s_t, a_t) - \lambda_1 \ell_{\texttt{acc}}(s_t, a_t)$, which concerns accuracy but ignores the fairness violation.

- $r_{\texttt{all}} = R_{\texttt{base}}(s_t, a_t) - \lambda_1 \ell_{\texttt{acc}}(s_t, a_t) - \lambda_2 \ell_{\texttt{fair}}(s_t, a_t)$, which considers both accuracy and fairness.

In particular, we choose non-zero $\lambda$ values by performing a grid search over an arithmetic sequence (i.e., 0.1 to 0.9, spaced 0.1 apart). We set the policy studied in Zhang et al. (2020) which myopically maximizes instantaneous utility as the baseline. For each RL approach, we run 80,000 time steps to ensure its convergence. Note that we do not include existing models for long-term fairness as they are not comparable. The hyperparameters we used are provided in the Appendix.

Although both the state transition and the threshold policy are defined at the group level, our RL sampling process still operates on individual-level samples. During each training iteration, we sample individual instances from the population distribution and apply the corresponding group-specific threshold to determine the decision outcomes. These instance-level decisions are used to compute the reward and optimize the policy. The group-level qualification distribution, which governs the state transitions, is updated once per epoch based on the aggregated outcomes of the selected individuals within each group. This separation ensures that the model captures long-term group-level effects while maintaining compatibility with standard instance-based sampling practices in reinforcement learning.

| $t_{00}$ | $t_{01}$ | $\alpha_a^*$ | $\alpha_b^*$ | $r_{\texttt{base}}$ ($\downarrow$) |
|------|------|-------|-------|-------|
| 0.1 | 0.1 | 0.100 | 0.100 | 0.000 |
| 0.1 | 0.5 | 0.167 | 0.167 | 0.000 |
| 0.1 | 0.9 | 0.500 | 0.500 | 0.000 |
| 0.5 | 0.1 | 0.357 | 0.357 | 0.000 |
| 0.5 | 0.5 | 0.391 | 0.523 | **0.132** |
| 0.5 | 0.9 | 0.833 | 0.833 | 0.000 |
| 0.9 | 0.1 | 0.500 | 0.500 | 0.000 |
| 0.9 | 0.5 | 0.643 | 0.643 | 0.000 |
| 0.9 | 0.9 | 0.391 | 0.523 | **0.132** |

Table 1: Results of $r_{\texttt{base}}$ on COMPAS dataset.

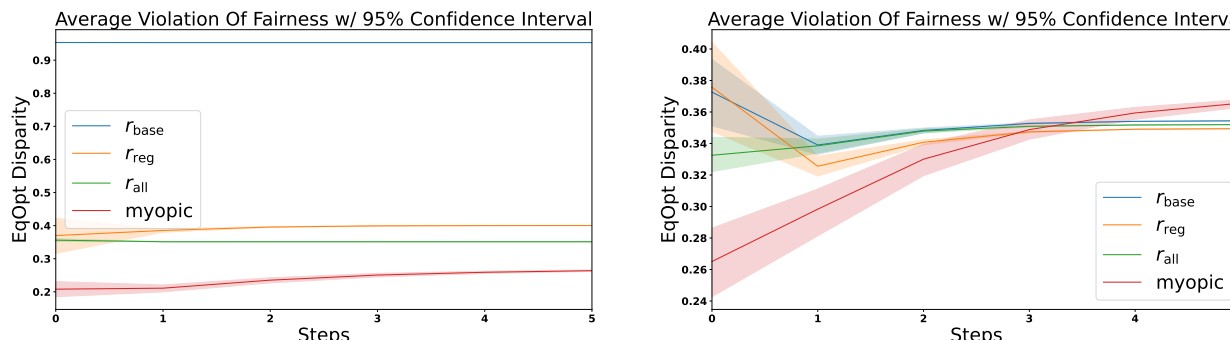

Figure 3: Violation of `EqOpt` for different policies on synthetic Gaussian data (left) and FICO dataset (right) at each time step.

## 6.2 Results

**RL is effective at achieving zero deterioration in the long run**  Fig 1 shows the dynamic of the qualification rate during a 20-step interplay between algorithmic decision and data distribution, where different trajectories (red lines with arrows) correspond to different initial qualification rates. Dots on the black diagonal indicate equal qualification rates across two groups. We have the following observations: (1) For both synthetic Gaussian and FICO datasets, RL algorithms with reward function $r_{\texttt{base}}$ can all effectively drive the qualification rates of two groups to be equal (the diagonal), *i.e.*, no deterioration between utility and fairness exists in the long run. (2) However, for myopic policy, the qualification rate of one group is significantly larger than the other group when the RL converges, suggesting the existence of the fairness-utility trade-off. Indeed, the heatmap shows the optimal utility subject to `EqOpt` gained by myopic policy is less than $r_{\texttt{base}}$, $r_{\texttt{reg}}$ and $r_{\texttt{all}}$.

**Add regularization to encourage higher utility.** The third and fourth columns of Fig 1 show the dynamic of the qualification rate with the regularized reward function. We observe that using $r_{\texttt{reg}}$ and $r_{\texttt{all}}$ as a reward achieves equilibrium with higher qualification rates than $r_{\texttt{base}}$ and myopic policy, as the equilibrium located is located closer to the top right corner in Fig. 1.

**Add regularization to encourage lower fairness violation.** Fig 3 shows the fairness violation of `EqOpt`, averaged over a 10-split mesh grid on $[0,1]^2$ as initialization of $\alpha_a, \alpha_b$ for five steps with 95% confidence interval, for different policies on synthetic Gaussian and FICO datasets at each step. For synthetic Gaussian data, RL agent trained with reward function $r_{\texttt{base}}$ constantly uses a policy that suffers a large violation of `EqOpt`, while policies with $r_{\texttt{reg}}$ violate much less. The fairness violation in policies with $r_{\texttt{all}}$ further decreases. However, they all suffer larger fairness violations than the myopic policy. We believe this is because although policies chosen by $r_{\texttt{reg}}$ are aware of fairness violations, the qualification rate parts dominate, i.e. the policy chooses to sacrifice fairness to achieve equalized and large qualification rates. For the FICO dataset, policies with rewards $r_{\texttt{base}}$, $r_{\texttt{acc}}$ and $r_{\texttt{fair}}$ suffer similar violations after converging, and all of these policies

achieve smaller violations than the myopic policy, because the distribution of FICO dataset only admits one equilibrium.

### 6.3 Discussion

**Equilibrium of qualification rate.** We are interested in understanding whether the qualification rate for each demographic group will converge or fluctuate once the reinforcement learning (RL) agent drives the population to a state with zero utility deterioration. Formally, we say the population distribution reaches *equilibrium* if the qualification rate of every demographic group converges, i.e. $\lim_{t\to\infty} \alpha_g = \alpha_0, \forall g \in \{a, b\}$. We compute the average dynamic (averaged over a 10-split mesh grid on $[0, 1]^2$ as initialization of $\alpha_a$, $\alpha_b$) for 5 steps with 95% confidence interval and show the result in Fig 2. We observe that RL agents with reward function $r_{\texttt{base}}, r_{\texttt{reg}}, r_{\texttt{all}}$ successfully choose policies that achieve equilibrium with equalized qualification rates, while the myopic policy attains equilibrium with a large gap between $\alpha_a$ and $\alpha_b$. Furthermore, the equilibrium of myopic policy is achieved more slowly compared to the RL policies.

**Effect of transition probabilities.** We provide an empirical analysis to show how the RL algorithm will perform under different dynamics (specified by different transition probabilities). Specifically, we set transition probabilities in the following way: $T_{00}^0 = T_{00}^1 = t_{00}, T_{01}^0 = T_{01}^1 = t_{00} \times \mathrm{tr}, T_{10}^0 = T_{10}^1 = t_{10}, T_{11}^0 = T_{11}^1 = t_{10} \times \mathrm{tr}$, with different configurations of $t_{00}$ and $t_{10}$ and a fixed $tr = 0.8$ (we do not observe difference on performance when altering tr). We train PPO policies for each setting, and show $r_{\texttt{base}}$ on COMPAS data in Table 1. It shows the qualification rates of the two groups and their absolute difference (smaller is better) to which the model converges. We observed that there were only two cases (highlighted in bold) where the policy failed to achieve equal qualification rates. This failure arguably occurred because the underlying MDP does not admit a stationary distribution where the qualification rates for group a ($\alpha_a$) and group b ($\alpha_b$) are equal at any time step. It is still possible that we may reach an unstable equilibrium in an intermediate state. However, we can only observe the failure after the deployment since the environment will be unknown to the RL policy.

## 7 Limitations and Future Work

Our current work only examines the overall utility deterioration for the entire population. However, policy-makers would likely be more interested in identifying the specific individuals whose utility has worsened due to enforcing fairness constraints. This phenomenon is also known as infra-marginality (Biswas et al., 2019) in the context of dynamic learning. Additionally, it is common for the qualification rate to decrease when fairness constraints are imposed, leading to user churn. We anticipate that future work could extend the analysis of utility deterioration to provide a more nuanced understanding at the group or individual level.

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

# A  Omitted Proofs

## A.1  Proof of Lemma 1

**Lemma 1.** *Assuming that the feature is well-behaved, there exist two threshold pairs $(\mu_a, \mu_b)$, $(\nu_a, \nu_b)$, such that*

$$\pi_a^*(x) := \mathbb{1}(x \geq \mu_a), \quad \pi_b^*(x) := \mathbb{1}(x \geq \mu_b)$$

*is the optimal policy pair for the unconstrained optimization problem Eqn (4); and*

$$\pi_a^\diamond(x) := \mathbb{1}(x \geq \nu_a), \quad \pi_a^\diamond(x) := \mathbb{1}(x \geq \nu_b)$$

*is the optimal policy pair for the fairness-constrained optimization problem defined in Eqn (5).*

*Proof.* In the first part of the proof, we show that the threshold policy is optimal for the unconstrained problem. It is obvious to see that maximizing the pair of $(\pi_a, \pi_b)$ to the total utility

$$\max_\pi \mathrm{Util}(\pi) = \max_{\pi_a, \pi_b} \left( p_a \mathbb{E}[\mathrm{Util}(\pi_a) \mid G = a] + p_b \mathbb{E}[\mathrm{Util}(\pi_b) \mid G = b] \right)$$

is equivalent to maximizing the group-specific utility for each group

$$\forall g \in \{a, b\}, \qquad \max_{\pi_g} \mathbb{E}[\mathrm{Util}(\pi_g) \mid G = g].$$

Then we expand the group-specific utility as

$$
\begin{aligned}
\mathbb{E}[\mathrm{Util}(\pi_g) \mid G = g] &= \sum_{\substack{i \in \{0,1\} \\ j \in \{0,1\}}} c_{ij} \mathbb{P}(\hat{Y} = i, Y = j \mid G = g) \\
&= \sum_{\substack{i \in \{0,1\} \\ j \in \{0,1\}}} c_{ij} \int_x \mathbb{P}(\hat{Y} = i, Y = j, X = x \mid G = g) dx \\
&= \int_x \mathbb{P}(X = x \mid G = g) \sum_{\substack{i \in \{0,1\} \\ j \in \{0,1\}}} c_{ij} \mathbb{P}(\hat{Y} = i, Y = j \mid X = x, G = g) dx
\end{aligned}
$$

(25)

Recall that $\pi_g(x) = \mathbb{P}(\hat{Y} = i \mid X = x, G = g)$ and $\gamma_g(x) = \mathbb{P}(Y = i \mid X = x, G = g)$. Since $Y \perp\!\!\!\perp \hat{Y}$ given the feature $X$ and group membership $G$,

$$
\begin{aligned}
&\sum_{\substack{i \in \{0,1\} \\ j \in \{0,1\}}} c_{ij} \mathbb{P}(\hat{Y} = i, Y = j \mid X = x, G = g) \\
&= \sum_{\substack{i \in \{0,1\} \\ j \in \{0,1\}}} \mathbb{P}(\hat{Y} = i \mid X = x, G = g) \mathbb{P}(Y = j \mid X = x, G = g) \\
&= c_{11} \pi_s(x) \gamma^s(x) + c_{10} \pi_s(x) \left(1 - \gamma_s(x)\right) + c_{01}(1 - \pi_s(x)) \gamma^s(x) + c_{00}(1 - \pi^s(x))\left(1 - \gamma^s(x)\right) \\
&= c_{00} + \left(c_{01} - c_{00}\right) \gamma_g(x) + \left((c_{11} - c_{10} - c_{01} + c_{00}) \gamma_g(x) + c_{10} - c_{00}\right) \pi_g(x)
\end{aligned}
$$

(26)

Combining Eqn (25) and Eqn (26), we have

$$
\begin{aligned}
&\mathbb{E}[\mathrm{Util}(\pi_g) \mid G = g] = \\
&\qquad \mathbb{E}_{X=x \mid G=g} \left[ c_{00} + \left(c_{01} - c_{00}\right) \gamma_g(x) + \left((c_{11} - c_{10} - c_{01} + c_{00}) \gamma_g(x) + c_{10} - c_{00}\right) \pi_g(x) \right]
\end{aligned}
$$

(27)

Note that $\forall x, \pi_g(X) \in [0, 1]$. To maximize the above quantity, we can always assign

$$\pi_g^*(x) = \begin{cases} 0, & \text{if } \gamma_g(x)(c_{11} - c_{10} - c_{01} + c_{00}) + c_{10} - c_{00} < 0 \\ 1, & \text{otherwise} \end{cases} \tag{28}$$

In other words, $\pi_g^*(x) = \mathbb{1}((c_{11} - c_{10} - c_{01} + c_{00})\gamma_g(x) + c_{10} - c_{00} > 0)$ is the optimal policy. Due to the monotonicity of $\gamma_g(x)$, $\pi_g^*(x) = \mathbb{1}\left(x \geq \gamma_g^{-1}(\frac{c_{00} - c_{10}}{c_{11} - c_{10} - c_{01} + c_{00}})\right)$ is a threshold policy.

To show that threshold policy is also optimal for the constrained optimization problem, it suffices to show that for an arbitrary pair of policy $(\pi_a^\diamond, \pi_b^\diamond)$, there always exists a pair of threshold policy $(\hat{\pi}_a^\diamond, \hat{\pi}_b^\diamond)$, where $\forall g, \hat{\pi}_g^\diamond = \mathbb{1}(x \geq \nu_g)$, such that $(\hat{\pi}_a^\diamond, \hat{\pi}_b^\diamond)$ also satisfies the fairness constraint, and the utility gained from threshold policy pair $(\hat{\pi}_a^\diamond, \hat{\pi}_b^\diamond)$ is greater than or equal to the utility gained by $(\pi_a^\diamond, \pi_b^\diamond)$.

First, following the proof of Lemma 1 in (Zhang et al., 2020), we can show that there always exists a pair of threshold policies $(\hat{\pi}_a^\diamond, \hat{\pi}_b^\diamond)$ that shares the same fairness metric as $(\pi_a^\diamond, \pi_b^\diamond)$. Then we may compute the difference of the corresponding group-specific utility for group $g$ by

$$\text{Util}(\hat{\pi}_g^\diamond) - \text{Util}(\pi_g^\diamond) = \mathbb{E}_{X=x|S=s}\left[(\hat{\pi}_g^\diamond(x) - \pi_g^\diamond(x))\left((c_{11} - c_{10} - c_{01} + c_{00})\gamma_g(x) + c_{10} - c_{00}\right)\right]$$
$$= \int_{\nu_g}^{\infty} \left(1 - \pi_g^\diamond(x)\right) f(x, g)dx - \int_{-\infty}^{\nu_g} \pi_g^\diamond(x) f(x, g)dx \tag{29}$$

where

$$f(x, g) = \left((c_{11} - c_{10} - c_{01} + c_{00})\gamma_g(x) + c_{10} - c_{00}\right)\mathbb{P}(X = x \mid G = g). \tag{30}$$

Let $\mathcal{P}_g^{\mathcal{C}}(x)$ denote the distribution related to the fairness metric, where

$$\mathcal{P}_g^{\text{DP}}(x) = \mathbb{P}(X = x \mid G = g) \tag{31}$$
$$\mathcal{P}_g^{\text{EqOpt}}(x) = \mathbb{P}(X = x \mid Y = 1, G = g) \tag{32}$$

Since both the arbitrary pair of policies and the threshold pair of policies satisfy the fairness constraint, i.e. $\mathbb{E}_{X \sim \mathcal{P}_g^{\mathcal{C}}}\left[\pi_g^\diamond(X)\right] = \mathbb{E}_{X \sim \mathcal{P}_g^{\mathcal{C}}}\left[\hat{\pi}_g^\diamond(X)\right]$, we have

$$\int_{\nu_g}^{\infty} \left(1 - \pi_g^\diamond(x)\right) \mathcal{P}_g^{\mathcal{C}}(x)dx = \int_{-\infty}^{\nu_g} \pi_g^\diamond(x) \mathcal{P}_g^{\mathcal{C}}(x)dx \tag{33}$$

Next, we show that $f(x, g)/\mathcal{P}_g^{\mathcal{C}}(x)$ is non-decreasing. For DP, it is obvious that

$$\frac{f(x, g)}{\mathcal{P}_g^{\text{DP}}(x)} = \left((c_{11} - c_{10} - c_{01} + c_{00})\gamma_g(x) + c_{10} - c_{00}\right)\frac{\mathbb{P}(X = x \mid G = g)}{\mathbb{P}(X = x \mid G = g)}$$
$$= (c_{11} - c_{10} - c_{01} + c_{00})\gamma_g(x) + c_{10} - c_{00} \tag{34}$$

is strictly increasing due to the monotonicity of $\gamma_g(x)$. For EqOpt,

$$\frac{f(x, g)}{\mathcal{P}_g^{\text{EqOpt}}(x)} = \left((c_{11} - c_{10})\gamma_g(x) - (c_{00} - c_{01})(1 - \gamma_g(x))\right)\frac{\mathbb{P}(X = x \mid G = g)}{\mathbb{P}(X = x \mid Y = 1, G = g)}$$
$$= (c_{11} - c_{10})\frac{\mathbb{P}(Y = 1 \mid X = x, G = g)\mathbb{P}(X = x \mid G = g)}{\mathbb{P}(X = x \mid Y = 1, G = g)}$$
$$- (c_{00} - c_{01})\frac{\mathbb{P}(Y = 0 \mid X = x, G = g)\mathbb{P}(X = x \mid G = g)}{\mathbb{P}(X = x \mid Y = 1, G = g)}$$
$$= (c_{11} - c_{10})\mathbb{P}(Y = 1 \mid G = g)$$
$$- (c_{00} - c_{01})\mathbb{P}(Y = 0 \mid G = g)\frac{\mathbb{P}(X = x \mid Y = 0, G = g)}{\mathbb{P}(X = x \mid Y = 1, G = g)} \tag{35}$$

Based on Definition 4, $\frac{\mathbb{P}(X=x|Y=0,G=g)}{\mathbb{P}(X=x|Y=1,G=g)}$ is monotonically decreasing, and the negative sign makes the likelihood ratio strictly increasing. Thus we conclude that $f(x,g)/\mathcal{P}_g^{\mathcal{C}}(x)$ is strictly increasing for both DP and EqOpt constraints.

Finally, we have

$$\int_{-\infty}^{\nu_g} \pi_g^{\diamond}(x) f(x,g) dx = \int_{-\infty}^{\nu_g} \pi_g^{\diamond}(x) \frac{f(x,g)}{\mathcal{P}_g^{\mathcal{C}}(x)} \mathcal{P}_{\mathcal{C}}^s(x) dx \tag{36}$$

$$\leq \int_{-\infty}^{\nu_g} \pi_g^{\diamond}(x) \frac{f(\nu_g,g)}{\mathcal{P}_g^{\mathcal{C}}(\nu_g)} \mathcal{P}_{\mathcal{C}}^s(x) dx \tag{37}$$

$$= \int_{\nu_g}^{\infty} (1 - \pi_g^{\diamond}(x)) \frac{f(\nu_g,g)}{\mathcal{P}_g^{\mathcal{C}}(\nu_g)} \mathcal{P}_{\mathcal{C}}^s(x) dx \tag{38}$$

$$\leq \int_{\nu_g}^{\infty} (1 - \pi_g^{\diamond}(x)) \frac{f(x,g)}{\mathcal{P}_g^{\mathcal{C}}(x)} \mathcal{P}_{\mathcal{C}}^s(x) dx \tag{39}$$

$$= \int_{\nu_g}^{\infty} (1 - \pi_g^{\diamond}(x)) f(x,g) dx \tag{40}$$

The above inequalities are due to the monotonicity of $f(x,g)/\mathcal{P}_g^{\mathcal{C}}(x)$. Therefore,

$$\forall g \in \{a,b\}, \quad \text{Util}(\hat{\pi}_g^{\diamond}) - \text{Util}(\pi_g^{\diamond}) = \int_{\nu_g}^{\infty} \left(1 - \pi_g^{\diamond}(x)\right) f(x,g) dx - \int_{-\infty}^{\nu_g} \pi_g^{\diamond}(x) f(x,g) dx \geq 0.$$

We finish the proof. □

## A.2  Proof of Lemma 2

**Lemma 2.** *For any group $g$, the optimal threshold $\mu_g$ for unconstrained optimization problem satisfies $\gamma_g(\mu_g) = \gamma^*$, where*

$$\gamma^* = \frac{c_{00} - c_{10}}{c_{00} + c_{11} - c_{10} - c_{01}}. \tag{41}$$

*The optimal threshold pair $(\nu_a, \nu_b)$ for the constrained optimization problem satisfies*

$$\sum_g p_g \left(\gamma_g(\nu_g) - \gamma^*\right) = 0, \quad \text{or} \tag{DP constrained}$$

$$\sum_g p_g \left(1 - \frac{\gamma^*}{\gamma_g(\nu_g)}\right) \mathbb{P}(Y = 1 \mid G = g) = 0 \tag{EqOpt constrained}$$

*where $p_g = \mathbb{P}(G = g)$ represents the population distribution.*

*Proof.* In the proof of Lemma 1, we already have the optimal threshold policy for the unconstrained problem:

$$\forall g, \quad \pi_g^*(x) = \mathbb{1}(x \geq \mu_g)$$
$$\text{where} \quad \mu_g = \gamma_g^{-1}\left(\frac{c_{00} - c_{10}}{c_{00} + c_{11} - c_{10} - c_{01}}\right) \tag{42}$$

Thus we can easily get $\gamma(\mu_g) = \gamma^*$.

Next, the parity constraint of the fair threshold policy $\mathbb{E}_{X \sim \mathcal{P}_a^{\mathcal{C}}}[\pi_a^{\diamond}(X)] = \mathbb{E}_{X \sim \mathcal{P}_b^{\mathcal{C}}}[\pi_b^{\diamond}(X)]$ implies

$$\int_{\nu_a}^{\infty} \mathcal{P}_g^{\mathcal{C}}(x) dx = \int_{\nu_b}^{\infty} \mathcal{P}_b^{\mathcal{C}}(x) dx. \tag{43}$$

Taking the derivative simultaneously, we have

$$\mathcal{P}_a^{\mathcal{C}}(\nu_a) d\nu_a = \mathcal{P}_b^{\mathcal{C}}(\nu_b) d\nu_b. \tag{44}$$

The utility of optimal fair policy is

$$
\text{Util}(\pi^\diamond) = \sum_{g \in \{a,b\}} \Big( \mathbb{E}_{X=x|G=g} \left[ c_{00} + (c_{01} - c_{00}) \gamma_g(x) \right]
$$

$$
+ \mathbb{E}_{X=x|G=g} \big[ ((c_{11} - c_{10} - c_{01} + c_{00}) \gamma_g(x) + c_{10} - c_{00}) \pi_g(x) \big] \Big) \quad (45)
$$

Note that the first component inside the expectation is a constant. Taking the derivative of the above equation with respect to the threshold $\nu_a$, we have

$$
\frac{d}{d\nu_a} \text{Util}(\pi^\diamond) = \frac{d}{d\nu_a} \sum_{g \in \{a,b\}} \mathbb{E}_{X=x|G=g} \big[ ((c_{11} - c_{10} - c_{01} + c_{00}) \gamma_g(x) + c_{10} - c_{00}) \pi_g^\diamond(x) \big]
$$

$$
= \frac{d}{d\nu_a} \left( p_a \int_{\nu_a}^{\infty} f(x,a) dx + p_b \int_{\nu_b}^{\infty} f(x,a) dx \right)
$$

$$
= p_a \frac{d}{d\nu_a} \int_{\nu_a}^{\infty} f(x,a) dx + p_b \frac{d\nu_b}{d\nu_a} \frac{d}{d\nu_b} \int_{\nu_b}^{\infty} f(x,a) dx
$$

$$
= -p_a f(\nu_a, a) - p_b \frac{d\nu_b}{d\nu_a} f(\nu_b, b) \quad (46)
$$

where $f(\cdot, \cdot)$ is defined in Eqn (30). The optimal fair policy satisfies that $\frac{d}{d\nu_a} \text{Util}(\pi^\diamond) = 0$. Combining Eqn (44) and Eqn (46), we have

$$
\sum_{g \in \{a,b\}} p_g \frac{f(\nu_g, g)}{\mathcal{P}_g^{\mathcal{C}}(\nu_g)} = 0 \quad (47)
$$

Substituting Eqn (31) and Eqn (32) respectively, we have for `DP`,

$$
\sum_g p_g \left( (c_{00} + c_{11} - c_{01} - c_{10}) \gamma_g(\nu_g) - (c_{00} - c_{10}) \right) = 0. \quad (48)
$$

Or equivalently,

$$
\sum_g p_g \left( \gamma_g(\nu_g) - \frac{c_{00} - c_{10}}{c_{00} + c_{11} - c_{01} - c_{10}} \right) = 0. \quad (49)
$$

Similarly for `EqOpt`, we have

$$
\sum_g p_g \left( \gamma_g(\nu_g) - \frac{c_{00} - c_{10}}{c_{00} + c_{11} - c_{01} - c_{10}} \right) \frac{\mathbb{P}(X = \nu_g \mid G = g)}{\mathbb{P}(X = \nu_g \mid Y = 1, G = g)} = 0. \quad (50)
$$

By Bayes' rule and the definition of $\gamma_g(\cdot)$, we may arrange the above equation as

$$
\sum_g p_g \left( 1 - \frac{\gamma^*}{\gamma_g(\nu_g)} \right) \mathbb{P}(Y = 1 \mid G = g) = 0. \quad (51)
$$

Complete the proof. $\qquad \square$

### A.3  Proof of Theorem 3

**Theorem 3.** *Suppose that the feature $X$ is both well-behaved and well-separated.*

*(a) For `DP`, there is no utility deterioration if and only if the qualification rates of the two groups are equal, i.e. $\mathbb{P}(Y = 1 \mid G = a) = \mathbb{P}(Y = 1 \mid G = b)$, if the following equation does not admit other solutions for the specific choices of population distributions of $\mathbb{P}(X = x \mid Y = 1, G = g)$ and cost matrix $c_{ij}$ :*

$$
\int_{\gamma_a^{-1}(\gamma^*)}^{\infty} \mathbb{P}(X = x \mid G = a) dx = \int_{\gamma_b^{-1}(\gamma^*)}^{\infty} \mathbb{P}(X = x \mid G = b) dx. \quad (52)
$$

*(b) For `EqOpt`, there is no utility deterioration if and only if the qualification rates of the two groups are equal, i.e. $\mathbb{P}(Y = 1 \mid G = a) = \mathbb{P}(Y = 1 \mid G = b)$.*

*Proof.* Taking the partial derivative of Eqn 14 with respect to the optimal fair threshold $\nu_g$, we have

$$\frac{\partial \Delta}{\partial \nu_g} = (c_{00} + c_{11} - c_{01} - c_{10})\left(\gamma_g(\nu_g) - \gamma_g(\mu_g)\right)\mathbb{P}(X = \nu_g \mid G = g) \tag{53}$$

where $\mu_g$ is the optimal threshold for unconstrained problem. When $\nu_g < \mu_g$, $\gamma_g(\nu_g) < \gamma_g(\mu_g)$ and $\Delta$ is decreasing; otherwise, $\gamma_g(\nu_g) > \gamma_g(\mu_g)$ and $\Delta$ is increasing. This monotonicity analysis tells us that $\Delta = 0$ if and only if $\forall g, \mu_g = \nu_g$. That being said, the optimal unconstrained policy also satisfies the fairness constraints. Let $\alpha_g = \mathbb{P}(Y = 1 \mid G = g)$ denote the group-specific qualification rate. Now we will prove $\alpha_g$ is constant across different group $g$ is the sufficient and necessary condition of $\mu_g = \nu_g$.

*Sufficient condition:* Suppose that $\alpha_g$ is the same for different group $g$. By Bayes' rule we have

$$\gamma_g(\mu_g) = \mathbb{P}(Y = 1 \mid X = \mu_g, G = g) = \frac{\mathbb{P}(X = \mu_g \mid Y = 1, G = g)}{\mathbb{P}(X = \mu_g \mid G = g)}\mathbb{P}(Y = 1 \mid G = g). \tag{54}$$

Then $\gamma_a(\mu_a) = \gamma_b(\mu_b)$ implies that $\mu_a = \mu_b$. By the law of total probability,

$$\begin{aligned}\mathbb{P}(X = x \mid G = g) &= \mathbb{P}(X = x \mid Y = 1, G = g)\mathbb{P}(Y = 1, G = g) \\ &\quad + \mathbb{P}(X = x \mid Y = 0, G = g)(1 - \mathbb{P}(Y = 1, G = g)) \\ &= \alpha_g\mathbb{P}(X = x \mid Y = 1, G = g) + (1 - \alpha_g)\mathbb{P}(X = x \mid Y = 0, G = g)\end{aligned} \tag{55}$$

Due to the separation assumption, $\mathbb{P}(X = x \mid G = g)$ remains the same across different group. By $\mu_a = \mu_b$, we can obtain that

$$\int_{\mu_a}^{\infty} \mathbb{P}(X = x \mid G = a)dx = \int_{\mu_b}^{\infty} \mathbb{P}(X = x \mid G = b)dx, \tag{56}$$

which means the optimal unconstrained policy also satisfies the `DP` constraint. Similarly, the parity constraint for `EqOpt` also holds:

$$\int_{\mu_a}^{\infty} \mathbb{P}(X = x \mid Y = 1, G = a)dx = \int_{\mu_b}^{\infty} \mathbb{P}(X = x \mid Y = 1, G = b)dx. \tag{57}$$

*Necessary condition:* Suppose that $\mu_g = \nu_g$. The parity constraints again show that $\mu_a = \mu_b$. For `DP`, it is straightforward to see that a universal solution of Eqn (56) is $\alpha_a = \alpha_b$ due to the fact that $\mu_a = \mu_b$ and $X$ is well-separated. It will depend on the specific choice of the distribution $\mathbb{P}(X = x \mid Y = 1, G = g)$ if Eqn (56) admits another solution $\alpha_a \neq \alpha_b$.

For `EqOpt`, by Bayes' rule we have

$$\gamma_g(\mu_g) = \mathbb{P}(Y = 1 \mid X = \mu_g, G = g) = \frac{\mathbb{P}(X = \mu_g \mid Y = 1, G = g)}{\mathbb{P}(X = \mu_g \mid G = g)}\mathbb{P}(Y = 1 \mid G = g) \tag{58}$$

If $\alpha_a \neq \alpha_b$, it is impossible that both $\mu_a = \mu_b$ and $\gamma_a(\mu_a) = \gamma_b(\mu_b)$ hold simultaneously. Then it contradicts with our previous solution $\gamma(\mu_a) = \gamma(\mu_b) = \gamma^*$. By contradiction, we conclude that $\alpha_a = \alpha_b$. $\qquad\square$

# B    More Experimental Results

## B.1    Computing Infrastructure

We conducted all the experiments on a server with four RTX A6000 GPUs. The average time cost for one run is roughly five minutes for finding the optimal policy.

## B.2 Experimental Results for DP

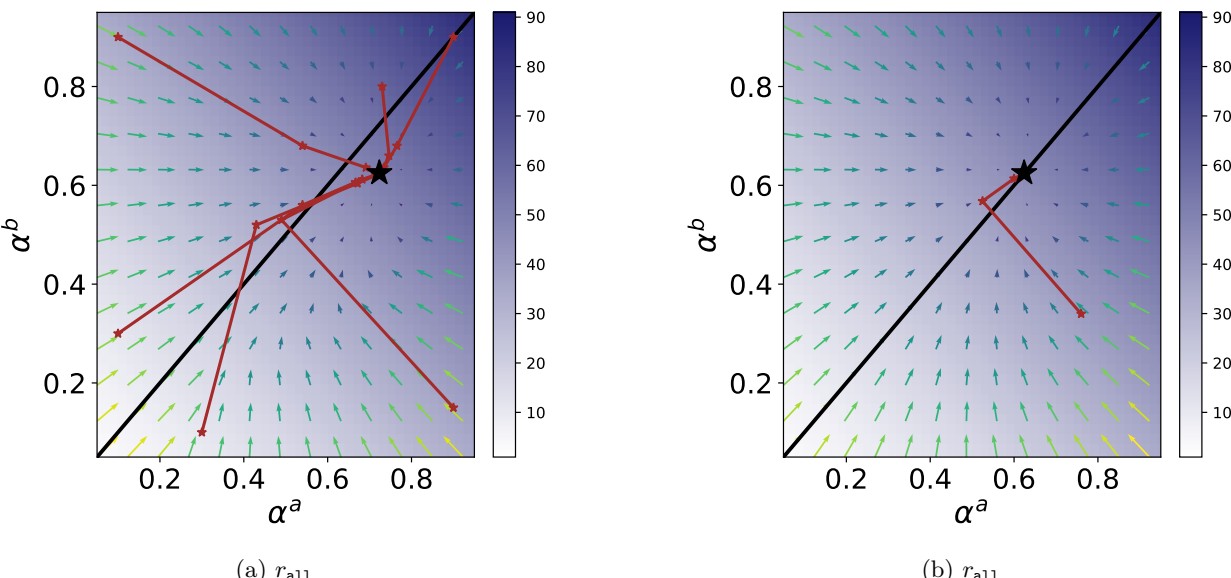

(a) $r_{\texttt{all}}$

(b) $r_{\texttt{all}}$

Figure 4: Trajectories for qualification rates with different initialization for synthetic Gaussian data (left) and the FICO dataset (right). For synthetic Gaussian data, the trajectories converge to a state with slightly different qualification rate, while the trajectories for FICO dataset is similar with EqOpt.

## C Hyperparameters for PPO

| | Value Considered | Value |
|---|---|---|
| Batch size | {8, 16, 32, 64, 128, 256, 512} | 8 |
| n_timesteps | {8e5} | 8e5 |
| n_steps | {8, 16, 32, 64, 128, 256, 512, 1024, 2048} | 256 |
| gamma | {0.9, 0.95, 0.98, 0.99, 0.995, 0.999, 0.9999} | 0.9 |
| learning_rate | float | 0.0052 |
| n_epochs | {1, 5, 10, 20} | 5 |
| activation | {ReLU, Tanh} | ReLU |
| net_architecture | {small, medium} | small |
| vf_coeficient | [0,1] | 0.2532 |
| max_grad_norm | {0.3, 0.5, 0.6, 0.7, 0.8, 0.9, 1, 2, 5} | 0.3 |
| gae_lambda: | {0.8, 0.9, 0.92, 0.95, 0.98, 0.99, 1.0} | 0.92 |

Table 2: Hyperparameters for PPO with reward function $r_{\text{base}}$ for synthetic Gaussian data.

|  | Value Considered | Value |
|---|---|---|
| Batch size | {8, 16, 32, 64, 128, 256, 512} | 512 |
| n_timesteps | {8e5} | 8e5 |
| n_steps | {8, 16, 32, 64, 128, 256, 512, 1024, 2048} | 16 |
| gamma | {0.9, 0.95, 0.98, 0.99, 0.995, 0.999, 0.9999} | 0.95 |
| learning_rate | float | 0.0012 |
| n_epochs | {1, 5, 10, 20} | 5 |
| activation | {ReLU, Tanh} | ReLU |
| net_architecture | {small, medium} | small |
| vf_coeficient | [0,1] | 0.5359 |
| max_grad_norm | {0.3, 0.5, 0.6, 0.7, 0.8, 0.9, 1, 2, 5} | 0.6 |
| gae_lambda: | {0.8, 0.9, 0.92, 0.95, 0.98, 0.99, 1.0} | 0.98 |

Table 3: Hyperparameters for PPO with reward function $r_{\texttt{reg}}$ for synthetic Gaussian data.

|  | Value Considered | Value |
|---|---|---|
| Batch size | {8, 16, 32, 64, 128, 256, 512} | 8 |
| n_timesteps | {8e5} | 8e5 |
| n_steps | {8, 16, 32, 64, 128, 256, 512, 1024, 2048} | 256 |
| gamma | {0.9, 0.95, 0.98, 0.99, 0.995, 0.999, 0.9999} | 0.9 |
| learning_rate | float | 0.0052 |
| n_epochs | {1, 5, 10, 20} | 5 |
| activation | {ReLU, Tanh} | ReLU |
| net_architecture | {small, medium} | small |
| vf_coeficient | [0,1] | 0.2532 |
| max_grad_norm | {0.3, 0.5, 0.6, 0.7, 0.8, 0.9, 1, 2, 5} | 0.3 |
| gae_lambda: | {0.8, 0.9, 0.92, 0.95, 0.98, 0.99, 1.0} | 0.92 |

Table 4: Hyperparameters for PPO with reward function $r_{\texttt{all}}$ for synthetic Gaussian data.

|  | Value Considered | Value |
|---|---|---|
| Batch size | {8, 16, 32, 64, 128, 256, 512} | 16 |
| n_timesteps | {8e5} | 8e5 |
| n_steps | {8, 16, 32, 64, 128, 256, 512, 1024, 2048} | 16 |
| gamma | {0.9, 0.95, 0.98, 0.99, 0.995, 0.999, 0.9999} | 0.95 |
| learning_rate | float | 0.0007 |
| n_epochs | {1, 5, 10, 20} | 10 |
| activation | {ReLU, Tanh} | ReLU |
| net_architecture | {small, medium} | small |
| vf_coeficient | [0,1] | 0.3346 |
| max_grad_norm | {0.3, 0.5, 0.6, 0.7, 0.8, 0.9, 1, 2, 5} | 0.9 |
| gae_lambda: | {0.8, 0.9, 0.92, 0.95, 0.98, 0.99, 1.0} | 0.98 |

Table 5: Hyperparameters for PPO with reward function $r_{\texttt{base}}$ for FICO dataset.

| | Value Considered | Value |
|---|---|---|
| Batch size | {8, 16, 32, 64, 128, 256, 512} | 256 |
| n_timesteps | {8e5} | 8e5 |
| n_steps | {8, 16, 32, 64, 128, 256, 512, 1024, 2048} | 16 |
| gamma | {0.9, 0.95, 0.98, 0.99, 0.995, 0.999, 0.9999} | 0.98 |
| learning_rate | float | 0.0007 |
| n_epochs | {1, 5, 10, 20} | 5 |
| activation | {ReLU, Tanh} | ReLU |
| net_architecture | {small, medium} | small |
| vf_coeficient | [0,1] | 0.4989 |
| max_grad_norm | {0.3, 0.5, 0.6, 0.7, 0.8, 0.9, 1, 2, 5} | 0.5 |
| gae_lambda: | {0.8, 0.9, 0.92, 0.95, 0.98, 0.99, 1.0} | 1.0 |

Table 6: Hyperparameters for PPO with reward function $r_{\texttt{reg}}$ for FICO dataset.

| | Value Considered | Value |
|---|---|---|
| Batch size | {8, 16, 32, 64, 128, 256, 512} | 8 |
| n_timesteps | {8e5} | 8e5 |
| n_steps | {8, 16, 32, 64, 128, 256, 512, 1024, 2048} | 256 |
| gamma | {0.9, 0.95, 0.98, 0.99, 0.995, 0.999, 0.9999} | 0.9 |
| learning_rate | float | 0.0052 |
| n_epochs | {1, 5, 10, 20} | 5 |
| activation | {ReLU, Tanh} | ReLU |
| net_architecture | {small, medium} | small |
| vf_coeficient | [0,1] | 0.2532 |
| max_grad_norm | {0.3, 0.5, 0.6, 0.7, 0.8, 0.9, 1, 2, 5} | 0.3 |
| gae_lambda: | {0.8, 0.9, 0.92, 0.95, 0.98, 0.99, 1.0} | 0.92 |

Table 7: Hyperparameters for PPO with reward function $r_{\texttt{all}}$ for FICO dataset.

