# OpenReview forum: "Long-Term Fairness Without Utility Deterioration"
_TMLR — Rejected by TMLR_

### Review · Reviewer_6Y3U · 2025-05-31

**Summary Of Contributions:**

The paper explores the trade-off between fairness and utility in dynamic learning environments, where the population distribution shifts over time in response to algorithmic decisions. The paper then proposes a reinforcement learning approach that incorporates a reward function encouraging equal qualification rates for learning in the dynamic environment. Empirical results on synthetic and real-world tabular datasets demonstrate that their method can drive the system towards this desired state.

**Audience:**

Yes

**Claims And Evidence:**

Yes

**Requested Changes:**

see weaknesses

**Strengths And Weaknesses:**

**Strengths**
- The paper rigorously derives a necessary and sufficient condition for zero utility deterioration under the dynamic learning setting, which is important and interesting.
- The use of MDPs and RL is appropriate for modeling sequential decision-making with population feedback.
- The paper provides empirical validation on synthetic and real-world tabular datasets.

**Weaknesses**
- Theorem 3 shows that achieving equal qualification rates across groups eliminates utility deterioration. However, this is essentially equivalent to enforcing DP, as stated in the definition of qualification rate in Eq. (1). This raises a critical question. If fairness is achieved by enforcing DP, then by definition the model would not experience a utility trade-off. But this result is tautological rather than insightful. It fails to address the inherent tension between fairness and utility in more realistic scenarios where equal qualification rates are not naturally achievable.

- The paper does not clearly specify what kinds of data shifts are considered in the dynamic environment. That is whether the feature distribution shifts, the label distribution changes, or both. Eq. (24) simply introduces the fairness regularization term into the RL reward without explaining how the system realistically models population adaptation in the long term.

- The experiments are confined to tabular datasets (synthetic, FICO, and COMPAS) and omit any image or high-dimensional data. Moreover, there is no thorough comparison to alternative methods. This makes it difficult to judge the practical effectiveness of the proposed approach compared to existing methods in the literature.

- The implementation essentially plugs the qualification rate equality constraint into the reward function of a standard RL algorithm. This straightforward modification in Eq. (24) does not constitute a fundamentally novel algorithmic contribution. The paper also does not compare to recent advances in long-term fairness with dynamic data distributions that might offer more substantial innovation.

---

> ### Author Response · Authors · 2025-07-15
> **Response to Reviewer 6Y3U**
>
> **Response to Weakness 1:**
> We thank the reviewer for this thoughtful question. We would like to respectfully clarify that enforcing demographic parity (DP) is not equivalent to achieving equal qualification rates across groups. Demographic parity pertains to the model’s decision outcomes - ensuring that the selection rates are equal across groups - whereas the qualification rate, as defined in Eq. (1), reflects the underlying distribution of qualified individuals in the population, independent of the model’s predictions.
>
> Theorem 3 establishes that when qualification rates become equal across groups, utility deterioration vanishes. However, this is not tautological, as it does not follow directly from applying a DP constraint. In fact, a model may satisfy demographic parity while still selecting individuals from groups with different underlying qualification distributions, resulting in disparate long-term utility effects.
> Moreover, our framework explicitly models population dynamics, wherein qualification rates evolve over time as a result of the model’s decisions. Therefore, equal qualification rates are not a static constraint, but rather an emergent property of the interaction between the decision-maker and users. The result in Theorem 3 highlights a key insight: long-term utility deterioration is inherently tied to the alignment of group-level qualification distributions, which may or may not align with fairness constraints like DP at any given point in time.
>
> **Response to Weakness 2:**
> We would like to remind the reviewer that we have clarified that we are focusing on the label shift in Section 5.
>
> **Response to Weakness 3:**
> We appreciate the reviewer’s point. Simulating population dynamics requires well-defined qualification outcomes and control over long-term feedback, which are not practical or meaningful in image settings. Tabular datasets allow us to explicitly model and measure these dynamics, making them appropriate for validating our theoretical framework.
>
> **Response to Weakness 4:**
> We appreciate the reviewer’s feedback. Our key contribution is not the RL algorithm itself but the theoretically grounded objective that targets long‑term utility under evolving qualification dynamics. Theorem 3 shows that aligning group qualification rates is both necessary and sufficient to eliminate utility deterioration over time. This insight directly motivates our reward shaping in Eq. (24), ensuring that policy updates account for the population‑level feedback loop. Although we leverage standard RL techniques for optimization, the novelty lies in formulating and analyzing a fairness criterion that guarantees preservation of utility in dynamic settings. We will clarify this emphasis in the revised manuscript.

---

> > ### Comment · Reviewer_6Y3U · 2025-08-01
> > **Official Comment by Reviewer 6Y3U**
> >
> > Thank you for your thoughtful responses. However, I remain concerned about the limitations of the experimental evaluation. The absence of comparisons with recent long-term fairness methods and the exclusive use of tabular datasets limit the assessment of the robustness and generalizability of the approach, especially for high-dimensional or real-world scenarios where fairness challenges are prominent.

---

> ### Comment · Reviewer_6Y3U · 2025-08-18
> **compared methods**
>
> The following recent works may serve as useful baselines or points of discussion to strengthen the experimental evaluation:
> 1. Yin et al., Long-Term Fairness with Unknown Dynamics (NeurIPS 2023).
> 2. Lear & Zhang, A Causal Lens for Learning Long-Term Fair Policies (ICLR 2025).
> 3. Ma & Xu, Promoting Fairness Among Dynamic Agents in Online-Matching Markets (NeurIPS 2024).
> 4. Deng et al., Reinforcement Learning with Stepwise Fairness Constraints (AISTATS 2023).

---

### Review · Reviewer_K9UY · 2025-06-10

**Summary Of Contributions:**

In this work, the authors study the tradeoff between utility and fairness. They examine two optimal decision policies: one that considers only utility and another that incorporates fairness constraints. These policies are used to define utility deterioration which is the absolute difference in utility between the two policies. The authors then identify conditions under which no utility deterioration occurs for two common fairness objectives: demographic parity and equal opportunity. The core theoretical contribution is a necessary and sufficient condition under which long-term fairness does not reduce utility: the equality of qualification rates across demographic groups. Building on this insight, they design reward functions for a Markov Decision Process (MDP) and conduct simulation experiments using both real and synthetic datasets.

**Audience:**

Yes

**Claims And Evidence:**

Yes

**Requested Changes:**

-	Please fix possible typos in equation (25) and figure 4 (tractories) in the appendix. Also (Previousous) in section 4.

-	It would be helpful if the authors could clarify the sampling procedure used during reinforcement learning. Specifically, it is unclear whether sampling is performed in batches or on a per-instance basis. The paper states that the action space consists of threshold policies defined per group, seemingly determined using group-specific qualification rates. In addition, the state transitions are defined at the group level, which suggests that the update may involve sampling multiple instances per group and then applying the corresponding thresholds. This departs from the typical RL setup where transitions are usually instance-based. Could the authors clarify this?

-	There are other works on long-term fairness such as [1,2]. I'm curious as to why these not comparable?

[1] Yu, Eric, et al. "Policy Optimization with Advantage Regularization for Long-Term Fairness in Decision Systems." Advances in Neural Information Processing Systems 35 (2022): 8211-8213.

[2] Hu, Yaowei, Jacob Lear, and Lu Zhang. "Striking a balance in fairness for dynamic systems through reinforcement learning." 2023 IEEE International Conference on Big Data (BigData). IEEE, 2023.

**Strengths And Weaknesses:**

Strength:

-	The paper provides a rigorous theoretical contribution by identifying a necessary and sufficient condition under which fairness constraints can be imposed without any utility deterioration in the long run. These results could be useful to others working in this area.

-	The experimental results indicate their reward function works to reduce the difference in qualification rates between groups, and thus utility deterioration. These findings validate the theoretical claim that long-term fairness can be achieved without compromising utility.

Weakness:

-	While the condition for achieving zero utility deterioration—namely, the equality of qualification rates across groups as established in Theorem 3—is theoretically reasonable, its implementation in Corollary 4 appears overly restrictive. Specifically, enforcing equal qualification rates at every timestep may impose unnecessary constraints on policy behavior, potentially compromising utility in the short term. Notably, Theorem 3 suggests that it is sufficient for qualification rates to converge over time, rather than remain equal throughout the trajectory. Therefore, it may be more appropriate to frame the fairness constraint in terms of a cumulative or long-horizon reward, rather than enforcing it instantaneously at each decision point. This relaxation could allow for more flexible policy learning while still ensuring convergence to the desired fairness equilibrium.

-	While many of the paper's assumptions are reasonable and theoretically grounded, the assumption that qualification rates are readily observable or accurately estimable by the policymaker is problematic in practice. In domains such as hiring, lending, or school admissions, it may be feasible to estimate qualification rates for the general population. However, obtaining reliable estimates for historically underserved or disadvantaged groups is considerably more challenging due to issues such as limited data availability, selection bias, and systemic underreporting. This limitation raises concerns about the practical viability and robustness of the proposed method in real-world deployments.

-	While the paper demonstrates the effectiveness of its proposed reward functions against baseline myopic policies, it does not compare performance with other long-term fairness approaches, such as those using replicator dynamics or causal models. Such a comparative study would have strengthened the empirical claims and helped contextualize the contribution relative to the current state of the art.

---

> ### Author Response · Authors · 2025-07-15
> **Response to Reviewer K9UY**
>
> **1. Typos:**
> We thank the reviewer for pointing out the typos in the manuscript. We have corrected the issues in the revised version.
>
> **2. Clarification of the Sampling Procedure in Reinforcement Learning:**
> We appreciate the reviewer’s thoughtful question regarding the sampling procedure and its connection to the group-level formulation. To clarify: although both the state transitions and threshold policies are defined at the group level, the sampling and learning process is conducted at the individual instance level. During each iteration of training, individual instances are sampled from the data distribution, and group-specific threshold policies are applied to determine individual outcomes. These outcomes are then aggregated to compute the group-level utility and update the group qualification distributions $\mathbf{\alpha}^g_t$. The state transitions are updated once per training epoch, based on the aggregated outcomes of selected individuals from each group. We have revised the manuscript to clarify this sampling process in RL training.
>
> **3. Comparison to Additional Long-Term Fairness Works:**
> We thank the reviewer for referencing additional relevant works on long-term fairness [Yu et al., 2022; Hu et al., 2023]. These methods offer valuable algorithmic contributions, particularly in reinforcement learning settings with dynamic feedback. However, our work focuses on a different modeling perspective, centered on the population-level qualification dynamics and theoretical characterization of utility deterioration over time. While the referenced works address fairness in sequential decision-making, our analysis explicitly formalizes how fairness constraints affect long-term utility through feedback on qualification distributions. This theoretical lens distinguishes our work and motivates the reward design and learning objectives used. We will update the related work section to better contextualize our contribution alongside these recent approaches.

---

### Review · Reviewer_gbuK · 2025-07-02

**Summary Of Contributions:**

The paper targets the trade-off between utility and fairness in machine learning. In particular, the goal is to ensure long-term fairness without compromising utility. The authors derive conditions under which this can be achieved and take those conditions as inspiration to develop reward functions for online reinforcement learning. The authors demonstrate the effectiveness of that reward function in synthetic and real-world datasets.

**Audience:**

Yes

**Claims And Evidence:**

Yes

**Requested Changes:**

1) Can you compare your reward function to some state-of-the-art reinforcement learning algorithm that considers long-term fairness?
2) Can you include the maximization problem already in the problem formulation?
3) Can you clarify the discussion around the loan example?
4) In the second paragraph of Section 4, "Previousius" should be "previous."

**Strengths And Weaknesses:**

Strengths
1) Without extensive background in the fairness literature, the problem setting appears relevant.
2) The paper is well-written.
3) Claims are supported by extensive proofs, although it could be mentioned in the main body that the proofs are in the appendix.

Weaknesses
1) There is some literature on fairness in RL, e.g., the survey by Jabbari et al. published at ICML 2017 summarizes some of it. Therefore, the comparison seems rather simple as the other reward functions do not account for fairness at all.
2) I found the discussion around the utility in (3) a bit confusing, as both the terms "utility" and "cost" are used. Intuitively, I would try to maximize utility but minimize cost. For the accuracy example the authors make, it seems that we want to maximize utility, in which case the definition of a cost seems counterintuitive. I think it should be part of the problem formulation to write down the optimization problem to make the goal clear.
3) I'm not quite following the first sentence of the second paragraph after Definition 3. It suffices for what? Also, for the example that follows, it is not clear to me what it should tell.

---

> ### Author Response · Authors · 2025-07-15
> **Response to Reviewer gbuK**
>
> **1. Comparison with state-of-the-art RL methods for long-term fairness:**
> We thank the reviewer for the suggestion to compare our reward function with existing reinforcement learning approaches that address fairness. A relevant example is the work by Jabbari et al. (2017), where the reward function is designed to promote fairness in actions, specifically by equalizing the probability of selecting each action across groups. Their notion of fairness is operationalized at the policy output level, focusing on parity in treatment. In contrast, our approach adopts a more outcome-oriented perspective. Our reward function is designed to minimize long-term utility deterioration, which reflects disparities in qualification outcomes that accumulate over time. Rather than enforcing parity in actions at each step, our framework encourages policies that steer the population toward equitable steady-state distributions. This distinction allows us to capture the delayed and compounding effects of decision-making, which are not addressed by action-level fairness objectives alone.
>
> **2. Inclusion of the maximization problem in the formulation:**
> We appreciate the reviewer’s comment. The optimization objective has been formally introduced in Eq. (4) and Eq. (5) in Section 4 of the paper.
>
> **3. Clarification of the loan example and scalar score assumption:**
> To clarify the sentence following Definition 3, we assume that the model maps multivariate inputs to a scalar qualification score. This assumption simplifies our theoretical analysis while preserving the essential structure of decision-making systems. The loan example illustrates how, in practice, complex financial and behavioral features are commonly summarized into a single scalar metric (e.g., a credit score), which then informs a threshold-based decision policy. This concrete example is meant to ground our modeling assumptions in a familiar and practical real-world context.
>
> **4. Typographical correction:**
> We thank the reviewer for pointing out the typographical error in Section 4. We have corrected the typos in the revised version.

---

> > ### Comment · Reviewer_gbuK · 2025-07-23
> >
> > Thank you for your replies. I would still think that it would help the paper to see empirically how the proposed techniques compare to other algorithms that focus on long-term fairness. I had seen that the maximization problem had been defined in Section 4. However, with that, the section titled "Problem Formulation" is missing an actual (mathematical) problem formulation.

---

### Review · Reviewer_xNTy · 2025-07-06

**Summary Of Contributions:**

The paper focuses on the trade off between utility (for example, some measure of accuracy) and fairness (between different demographic groups). It provides, under certain assumptions, conditions under which there is not utility deterioration if and only if qualification rates of different demographic groups are equal. It then moves beyond the static setting by examining utility deterioration in the long run. Based on the previous condition, it proposes reward functions for online reinforcement learning algorithms that allow for effective intervention---that is, small utility deterioration---in the long run. Experiments support the effectiveness of the approach in achieving vanishing utility deterioration.

**Audience:**

Yes

**Broader Impact Concerns:**

No concerns about the ethical implications of the work.

**Claims And Evidence:**

Yes

**Requested Changes:**

- Can the authors explain the theoretical guarantee of the algorithm propose in Section 5?

- There is a related paper in the game theory literature by. Zhang et al. called "Steering no-regret learners to a desired equilibrium." That paper has certain similarities with the goal of Section 5. Can the authors elaborate on that?

- Can the authors elaborate on the comparison with Dutta et al. (2020)?

- Some equations are missing punctuation marks.

- Typo in page 4: Previousous

- Try to avoid double parenthesis by adjusting the citation style.

- Use parentheses when the citation is syntantically not part of the sentence; for example, see the last sentence of page 8.

**Strengths And Weaknesses:**

Strengths: The paper studies an important and well-motivated problem in machine learning. Understanding conditions under which fairness is not at odds with utility is central problem, and the paper makes a concrete and, to my knowledge, new contribution in this line of work. Under two reasonable assumptions, it provides a sufficient and necessary condition characterizing utility deterioration. Perhaps the most interesting part of the paper concerns the dynamic setting. Here, they use an MDP formulation and show how to achieve vanishing utility deterioration in the limit by designing suitable interventions. This is a really interesting result that could have practical applications. Indeed, the experiments are quite thorough and demonstrate that the proposed approach is viable in several settings. Moreover, the paper overall is well written and organized and the key high-level ideas are well presented.

Weaknesses: From a theoretical standpoint, Section 5 does not contain any guarantee about having vanishing utility deterioration in the limit; I understand the implication of Corollary 4, but I would expect to see a formal guarantee about the qualification rates matching in the limit. Can the authors elaborate on that? I would expect to see further theoretical justification here.

---

> ### Author Response · Authors · 2025-07-15
> **Response to Reviewer xNTy**
>
> **Theoretical guarantee of the Algorithm in Section 5**
> We thank the reviewer for highlighting the importance of clarifying the theoretical implications of Corollary 4. We agree that a more explicit discussion of the convergence behavior of qualification rates under the demographic parity constraint is warranted.
> To elaborate, when a demographic parity constraint is imposed on the model prediction, the update dynamics become identical across groups. In this setting, each group’s qualification distribution​ evolves under a shared transition operator. Under standard assumptions on the transition matrix (e.g., ergodicity), these dynamics converge to a common stationary distribution. As a result, all groups will eventually share the same qualification rate, and any utility disparity or deterioration present at earlier iterations will asymptotically vanish. We have revised the manuscript to include this formal explanation in Section 5.
>
> **Comparison to Steering No-regret Learners to a Desired Equilibrium**
> We thank the reviewer for pointing out the connection to the work by Zhang et al. While Zhang et al.'s framework is rooted in online learning and game theory with no-regret learners, our formulation centers around population-level dynamics influenced by transition dynamics and fairness-constrained model predictions. In our case, the “steering” is induced implicitly by the fairness constraint (e.g., demographic parity), rather than by an external incentive mechanism. Nonetheless, the conceptual goal of guiding autonomous update processes to socially desirable equilibria is closely aligned.
>
> **Comparison to Duetta et al.**
> We thank the reviewer for pointing out the connection to Dutta et al. (2020). As clarified in the related work section, Dutta et al. focus on the fairness–accuracy trade-off in a static decision-making setting. In contrast, our work addresses the long-term utility deterioration that can arise when fairness constraints are imposed in a dynamic environment. We explicitly model this feedback loop and show how interventions aimed at short-term fairness can unintentionally reduce utility over time if population dynamics are not properly considered. Thus, while both works study fairness–utility trade-offs, our contribution lies in extending this analysis to a dynamic, temporally dependent setting where fairness interventions have downstream consequences on group-level outcomes.
>
> **Other Typos**
> Thank you for raising the attention to the typos and citation style. We have corrected the citation format.

---

### Decision · Action_Editor_SZXz · 2025-08-23

**Recommendation:** Reject

**Audience:**

Yes

**Audience Explanation:**

The lack of reasonable experiments and comparison with existing long-term fairness baselines diminishes the paper's relevance and interest to the community.

**Claims And Evidence:**

No

**Claims Explanation:**

This paper investigates the trade-off between fairness and utility in dynamic environments where a deployed model's policies can induce shifts in the population distribution over time. The authors model this interplay as a Markov Decision Process (MDP) and study  necessary and sufficient condition for achieving long-term fairness without any loss of utility. To leverage their findings, they propose novel reward functions for an online reinforcement learning framework. This framework is designed to guide a classifier-population system towards a desirable equilibrium where this identified condition is met.

While the reviewers appreciate the modeling and the overall framework, most reviewers lean toward rejection due to unresolved questions regarding the practical applicability of the work. A main criticism among the reviewers is the lack of comparison against relevant state-of-the-art long-term fairness algorithms (see the individual reviews for details). The experiments does not include recent fairness-aware baselines and are conducted exclusively on low-dimensional tabular datasets. This insufficient evaluation makes it impossible to assess the method's competitiveness, and reduces the community's interest in the work. Furthermore, reviewers raised concerns about the practical viability of the approach, regarding unrealistic assumption that group-specific qualification rates are readily and accurately observable. This amplifies the concern of the reviewers and further diminishes its interest to the community.

**Resubmission Of Major Revision:**

The authors may consider submitting a major revision at a later time.